# Dark Pigments in Entomopathogenic Fungal Microsclerotia: Preliminary Evidence of a 1,8-Dihydroxynaphthalene-melanin-like Compound in *Metarhizium robertsii*

**DOI:** 10.3390/jof9121162

**Published:** 2023-12-03

**Authors:** Daysi Espín-Sánchez, Lautaro Preisegger, Romina Mazzolenis, Marianela Santana, Mario C. N. Saparrat, Nicolás Pedrini, Carla Huarte-Bonnet

**Affiliations:** 1Instituto de Investigaciones Bioquímicas de La Plata “Profesor Doctor Rodolfo R. Brenner” (INIBIOLP), Universidad Nacional de La Plata (UNLP)-CCT-La Plata-CONICET, La Plata 1900, Argentina; daysi97espinsanchez@gmail.com (D.E.-S.); lautaropreisegger@gmail.com (L.P.); romi.mazzo@gmail.com (R.M.); marianelasantana@gmail.com (M.S.); 2Instituto de Fisiología Vegetal (INFIVE), Universidad Nacional de La Plata (UNLP)-CCT-La Plata-CONICET, La Plata 1900, Argentina; masaparrat@fcnym.unlp.edu.ar

**Keywords:** melanin inhibitors, tricyclazole, guaiacol, thermotolerance, oxidative stress

## Abstract

*Metarhizium robertsii* microsclerotia are fungal aggregates composed of compacted, pigmented hyphae. As they are highly tolerant to desiccation and produce infective conidia, they are promising candidates to be formulated as bioinsecticides. Despite this potential, the nature of the pigments within these structures remains unclear. In this study, routine culture media used for the differentiation of *M. robertsii* microsclerotia were supplemented with four melanin inhibitors, and the resulting propagules were characterized. Inhibitors of the 1,8-dihydroxynaphthalene (DHN)-melanin biosynthetic pathway such as tricyclazole and guaiacol induced significant phenotypic and molecular modifications in the obtained *M. robertsii* propagules, which exhibited a more spherical shape, reduced size, and increased susceptibility to desiccation, heat, and oxidative stress than microsclerotia obtained without inhibitors. Additionally, genes encoding for a polyketide synthase (Mrpks2) and a putative 1,3,6,8-tetrahydroxynaphthalene reductase (Mrthnr), potentially involved in the DHN-melanin biosynthetic pathway, were upregulated in fungi grown in the inhibitor-added media. In conclusion, *M. robertsii* microsclerotia contain melanins of type DHN that might play a role in both microsclerotia differentiation and environmental stress tolerance.

## 1. Introduction

Insect pathogenic fungi offer sustainable alternatives to chemical pesticides for pest control; however, their virulence and survival are often compromised by environmental conditions [1]. Fungal sclerotia are resilient overwintering propagules formed by the aggregates of pigmented hyphae. They are typically produced by different fungal taxa with diverse ecological roles, under either unfavorable environmental conditions or a lack of hosts [2,3]. They are mostly colored due to the synthesis and deposition of dark pigments known as melanins [4,5]. Smaller compact hyphal aggregates (50–600 μm) called microsclerotia (MS) [6,7,8,9] have been described in different entomopathogenic hypocrealean fungi such as *Metarhizium anisopliae* [8,10], *M. acridum*, *M. robertsii* [11], *M. brunneum* [12], *M. rileyi* [13], *Lecanicillium lecanii* [14], and *Beauveria bassiana* [15,16]. These propagules are promising candidates to use in biological control programs since they are tolerant to desiccation, UV, and oxidative stress and produce infective conidia [17]. Despite their potential, the nature of the pigments within these structures remains unclear.

Melanins are complex polymers mostly composed of aliphatic and indole or phenol-type aromatic structures [18]. They are secondary metabolites that show different coloration and heterogeneity regarding their composition, structure, and deposition pattern. From a functional point of view, melanin likely confers a survival advantage by protecting fungi against different growth-limiting factors such as UV radiation, low water potential (desiccation), and oxidative damage [19,20,21]. Fungal melanins are biosynthesized via different pathways, which involves a series of chemical transformations, including condensation and polymerization reactions [18]. Specifically, there are several different classes of fungal melanins based on the biochemical precursors, pathways, and/or intermediate metabolites involved. They can be γ-glutaminyl-3,4-dihydroxy-benzene (GDHB)-melanins, L-3,4-dihydroxyphenylalanine (DOPA)-melanins (which are also referred as eumelanins), 1,8-dihydroxynaphthalene (DHN)-melanins, catechol-melanins, pyomelanins, p-aminophenol (PAP)-melanins, green melanins, Asp-melanins, and 5-deoxybostrycoidin-based melanins [18,22]. In addition, a lot of other dark pigments in several fungal species, including a sort (type) called heterogenous melanins, have not yet been specified. Fungi may synthesize melanin using more than one pathway [18]. Inhibitors of putative pathways of melanin synthesis might help to elucidate these routes, and this can be used to infer the chemical nature of the pigment [23]. Bicyclopyrone is an inhibitor of 4-hydroxyphenylpyruvate dioxygenase, the enzyme responsible for the synthesis of homogentisate, preventing pyomelanin synthesis [24]. Kojic acid is a tyrosinase inhibitor, which can be used to block 3,4-dihydroxy phenylalanine (DOPA)-melanin synthesis [25]. There are two types of DHN-melanin synthesis inhibitors, which vary in their mechanism of action. While tricyclazole has inhibitory activity on the fungal reductases involved in the synthesis of 1,8-dihydroxynaphthalene, a key precursor of DHN-melanin, guaiacol blocks the polymerization reactions [26].

*Metarhizium* spp. are broad and versatile host range fungi, serving as arthropod pathogens, saprotrophs, and colonizers of both the rhizosphere and plant tissues [27]. The presence of a dark green pigment in the conidia of *Metarhizium* spp. has been associated with enhanced resistance to UV and heat stresses [28], and its non-melanin pigmentation biosynthetic pathway has been characterized [29]. However, no information is available regarding the pigments present in the microsclerotia from *Metarhizium* spp.

In this work, we studied the nature of dark pigments in MS from *M. robertsii* by adding different inhibitors of melanin biosynthesis during microsclerotial differentiation in liquid cultures. We found that tricyclazole and guaiacol altered fungal growth and microsclerotial differentiation, suggesting that the DHN pathway is involved in the pigmentation and environmental stress tolerance of MS from *M. robertsii*.

## 2. Materials and Methods

### 2.1. Fungal Isolate

The *M. robertsii* isolate ARSEF 2575 was obtained from the US Department of Agriculture (USDA)-ARS Collection of Entomopathogenic Fungal Cultures, Ithaca, NY, USA, and routinely cultured to obtain the conidia in potato dextrose agar (PDA, Biokar Diagnostics, Allonne, France) plates at 26 °C for 14 days.

### 2.2. Melanin Biosynthesis Inhibition

The basal medium previously described for the development of MS [30] was supplemented, once at a time, with four inhibitors of melanin synthesis: (i) bicyclopyrone (ACURON™ UNO, Syngenta, Buenos Aires, Argentina; [24]), (ii) kojic acid (5-hydroxy-2-(hydroxymethyl)-4-pyrone, Parafarm, Buenos Aires, Argentina), (iii) tricyclazole (5-methyl-1,2,4-triazolo[3,4-b] benzothiazole, BIM™, Corteva Agriscience, Montevideo, Uruguay), all of them at concentrations of 100, 350, 600, and 1200 ppm; and (iv) guaiacol (o-methoxyphenol, BDH Laboratory Reagents, Poole, England) at concentrations of 10, 35, 50, 75, and 100 ppm. These inhibitors target specific enzymes in the biosynthetic pathway of pyomelanin (i), DOPA-melanin (ii), or 1,8-dihydroxynaphthalene (DHN)-melanin (iii and iv). The experiment was performed in a completely randomized design with at least three replicate cultures per concentration and type of inhibitor tested, including a control that consisted of unamended basal medium. Ten milliliters of the conidial suspensions was prepared and adjusted to 5 × 10^7^ conidia mL^−1^ with 0.05% Tween 80 (Sigma-Aldrich, St. Louis, MO, USA) solution, vortexed, and inoculated into Erlenmeyer flasks (250 mL) containing 90 mL of either MS basal medium (control) or basal medium supplemented with the inhibitors mentioned. The cultures were set at 27 ± 1 °C in a rotary shaker incubator (CERTOMAT BS-1, Sartorius, Gottingen, Germany) at 250 rpm for 4 days. The experiments described below were repeated at least three times on different dates using the new fungal inoculum.

### 2.3. Propagule Characterization

One milliliter of 4-d-culture broth from each tested liquid media was diluted (1/10), and a drop was placed on glass slides (2.6 cm × 7.6 cm), covered with a coverslip (2.0 cm × 2.0 cm), and observed microscopically to count the propagules using a Nikon ECLIPSE E200 optical microscope (Nikon Corp., Tokio, Japan). The microscope imaging software (Labscope 4.0) was used to measure the aggregate sizes of at least 10 propagules of each of the four replicates. In the case of tricyclazole, additional concentrations (250–300 ppm) were tested for imaging analysis. At the end of the fermentation process, the fresh fungal biomass and culture broth were retrieved via centrifugation for 20 min at 7600× *g*. For each sample, the pH was measured from the culture broth (pH meter Altronix, Saen SRL, Buenos Aires, Argentina). The fungal biomass was then washed twice with sterile distiller water to remove the media components, and used for pigment quantification, tolerance assays, RNA extractions, and dry biomass measurements. The biomass measurements were adapted from [10] as follows: fungal cells harvested from 0.5 mL of culture broth were dried until they reached a constant weight, and the dried biomass production was further calculated as mg of dry mass per mL of culture broth.

Additionally, the MS harvested from 25 mL of either the basal medium (control) or basal medium supplemented with 350 ppm tricyclazole (350-tri) were similarly dried, weighed, and cultured on free-nutrient agar–water media (2% *w*/*v*) with ampicillin for 14 days at 26 °C. The fungal growth was monitored daily, and the conidia were harvested and suspended in sterile 0.01% Tween 80. Then, the conidial production was determined using a Neubauer chamber at 400× magnification under a phase contrast Primo Star microscope (Zeiss, Oberkochen, Germany), and expressed as the total conidia per gram of dry biomass.

### 2.4. Flaviolin and Melanin Quantification

The culture broth and fresh biomass were retrieved from cultures with or without 350 ppm tricyclazole as described before. The culture broth was used to determine the flaviolin production via spectrophotometric measurements at 340 nm according to [31]. The fresh biomass was washed, dried for 2 h at 80 °C, and ground in liquid nitrogen in a mortar and pestle. Subsequently, 100 mg of each sample were resuspended in 1M NaOH, and heated at 121 °C for 20 min [32]. The pigments were isolated and quantified via spectrophotometry at 459 nm according to [33], using the calibration curve method. The amount of dark pigments present in each sample was referred to as µg of melanin equivalent per gram of dry biomass.

### 2.5. Thermotolerance and Oxidative Stress Tolerance Assays

Four-day-old propagules harvested from either the control medium or medium supplemented with 350 ppm tricyclazole were washed and suspended in sterile distilled water. For the thermotolerance assays, the samples were adjusted to 100 propagules per mL, and 1 mL per sample was incubated for 0, 1, 3, and 5 h at 45 °C in a dry bath incubator. The control samples were maintained at 27 °C for the same time periods. Then, the samples were centrifuged and cultured in free-nutrient agar–water media (2% *w*/*v*) with ampicillin for 10 days at 27 °C. For the oxidative stress assays, each sample was cultured in free-nutrient agar–water media (2% *w*/*v*) with ampicillin supplemented with 0, 1, 4, 6, 10, 20, and 50 mM H_2_O_2_, and maintained for 10 days at 27 °C. After 10 days, the agar cultures for both assays were processed to assess the conidial production as described in [30].

### 2.6. Relative Gene Expression

The fresh biomass obtained from the control, 350-tri, and 75-guaiacol (75 ppm guaiacol) cultures was processed for total RNA extraction as described in [34]. Briefly, the total RNA was extracted by employing Tri Reagent (MRC, Cincinnati, OH, USA), including a DNA digestion step using a TURBO DNA-free kit (Thermo Fisher, Waltham, MA, USA). The RNA samples were quantified using a NanoDrop spectrophotometer (Thermo Fisher, Waltham, MA, USA), and the integrity was assessed on 1% (*w*/*v*) agarose gel. Two-step real-time polymerase chain reaction (RT-PCR) was carried out using the iScript cDNA synthesis kit, and iQ SYBR Green SuperMix (Bio-Rad, Hercules, CA, USA). Amplification was performed in AriaMx equipment (Agilent Genomics, Santa Clara, CA, USA) employing 20 ng reverse-transcribed total RNA for each sample. In order to confirm that only single products were amplified, a temperature-melting step was then performed. Four independent biological replicates were tested, with technical duplicates for each sample. The relative expression of genes encoding for a putative 1,3,6,8-tetrahydroxynaphthalene reductase (*Mrthnr*) and polyketide synthases (*Mrpk1* and *Mrpks2*) were studied, and gamma-tubuline (*Mrtub*) was used as the housekeeping gene. The *Mrthnr* sequence was obtained after BLASTP search using a 1,3,6,8-tetrahydroxynaphthalene reductase of *Aspergillus fumigatus* (the first reductase enzyme involved in the DHN biosynthesis pathway, susceptible to tricyclazole inhibition) as the query sequence. The oligonucleotides used are listed in Table 1. The relative expression ratio, the statistical analysis, and the expression plots were calculated using the REST software (version 2009, QIAGEN, Hilden, Germany).

### 2.7. Statistical Analysis

To evaluate the propagule parameters, differences among means were determined using analysis of variance (ANOVA), followed by the Tukey post-test. For the conidial production of dried biomass, pigment quantification, flaviolin accumulation, and differences between treatments at each exposure time in the both thermotolerance and oxidative stress assays, differences among means were determined using Student’s *t*-test. In all cases, GraphPad Prism software version 10 (GraphPad Software Inc., La Jolla, CA, USA) was used.

## 3. Results

### 3.1. Propagule Characterization

The growth of *M. robertsii* ARSEF 2575, and specifically the amount, size, and pigmentation of its microsclerotia, was strongly affected by the DHN pathway inhibitors tricyclazole and guaiacol being added to the culture media. However, no changes were observed with bicyclopyrone and kojic acid, inhibitors of the pyomelanin and DOPA pathways, respectively (Table 2). Supplementation with tricyclazole at a 600 ppm or higher concentration fully inhibited fungal growth, and the propagules derived from the cultures with sublethal concentrations showed several differences compared with the control cultures. Specifically, the propagules obtained with 350 ppm tricyclazole were morphologically different from the MS obtained in cultures without the inhibitor (Figure 1). The cumulates were more spherical (Figure 1) and significantly smaller (*p* < 0.01) than the control MS (Table 2). The dry biomass and propagule production were similar in cultures with low concentrations of tricyclazole (100–350 ppm), but the pH after 4 days of incubation was significantly lower (*p* < 0.01) than in the control cultures, and also than in the cultures with lethal concentrations (Table 2). Guaiacol also inhibits melanization via the DHN pathway, but further downstream than tricyclazole (see Section 4). Thus, we investigated the supplementation of this inhibitor in the culture medium. We found that 75 ppm guaiacol exhibits phenotypic characteristics like those observed with 350 ppm tricyclazole (350-tri). However, differences were also found compared with tricyclazole, i.e., there were no significant changes in pH, but there was a significant increase in propagule production compared to that obtained without supplementation with the inhibitor (MS). With this set of results, both MS and 350-tri cultures were selected to study the flaviolin accumulation in the remaining broth at the end of the fermentation, and conidial production, pigment concentration, and thermotolerance and oxidative stress tolerance of the fungal propagules.

### 3.2. Flaviolin and Melanin Quantification

Cultures of *M. robertsii* with 350-tri produced significantly more flaviolin than the control cultures (*p* < 0.0001) (Figure 2a). Pigment isolation and quantification from the MS and 350-tri cells showed that the MS contained twice the amount of pigments than the 350-tri-derived cells (*p* < 0.01) (Figure 2b).

### 3.3. Desiccation, Thermotolerance, and Oxidative Stress Tolerance Assays

Both the MS and 350-tri propagules were able to tolerate desiccation and produce viable conidia. However, the dry biomass produced by both propagules was significantly different (*p* < 0.05), i.e., the MS cultures produced 2.9 ± 0.3 × 10^10^ conidia/g, and the 350-tri propagules produced 1.0 ± 0.5 × 10^10^ conidia/g. Both propagules were also used to study heat tolerance, and the conidial production is shown in Figure 3a. For both the control and 350-tri propagules, a significant decrease in conidial production is observed following their incubation from 3 h treatment onward, indicating a reduced temperature tolerance after this period. Moreover, significant differences in conidial production were observed between the control and 350-tri propagules at the beginning (incubation control), 1 h, and 3 h, but not at 5 h treatment, when the values were notably lower for both conditions. Consequently, the treated propagules exhibit significantly lower conidial production than the controls (*p* < 0.0001) and a significantly reduced temperature tolerance after 3 h incubation (*p* < 0.01) (Figure 3a). Similarly, tolerance to oxidative stress was studied. Both the control and 350-tri propagules exhibit a concentration-dependent susceptibility to hydrogen peroxide, showing a significant decrease in conidial production as the concentration of hydrogen peroxide increases. In the case of the control samples, 100 mM inhibits the germination of the MS, whereas for 350-tri, inhibition occurs at 50 mM H_2_O_2_. Across all sublethal concentrations examined, there are significant differences in conidia production between the control and 350-tri groups (Figure 3b). Therefore, treated propagules are significantly less tolerant to oxidative stress than the MS.

### 3.4. Relative Gene Expression

The RNA extraction process for the samples treated with tricyclazole proved challenging due to the noticeable RNA degradation, potentially arising from the direct or indirect impact of the compound on the fungus. Consequently, we opted to work with biomass derived from the 75-guaiacol cultures, which exhibit a similar phenotype to that induced by 350-tricyclazole. The results concerning the relative expression of genes involved in the DHN-melanin synthesis pathway are illustrated in Figure 4. Notably, both the *Mrthnr* and *Mrpks2* genes exhibit a significant downregulation in the 75-guaiacol samples compared to the control samples, whereas *Mrpks1* shows an upregulated expression (*p* < 0.05).

## 4. Discussion

In this study, we report for the first time the effects of different inhibitors of melanin biosynthetic pathways on the *M. robertsii* ARSEF 2575 microsclerotia, and propose 1,8-dihydroxy naphthalene (DHN)-melanin as an active biosynthetic pathway. Most ascomycetous fungi use the polyketide pathway to synthesize the DHN types of melanins, 1,8-DHN being its main precursor [20]. Although some differences were reported in the fungal DHN-melanin pathway, in general, the initial step in this process involves the conversion of malonyl-CoA/Acetyl CoA into 1,3,6,8-tetrahydroxynaphthalene (1,3,6,8-THN) by polyketide synthases (PKSs). Following this, scytalone is generated by a reductase activity, and then its enzymatic dehydration leads to the formation of 1,3,8-trihydroxy naphthalene (1,3,8-THN), which is subsequently reduced into vermelone by a second reductase enzyme. Scytalone dehydratase further catalyzes another dehydration step, resulting in the intermediate 1,8-dihydroxy naphthalene (DHN). Finally, a laccase enzyme facilitates the dimerization and subsequent polymerization of the molecules to produce DHN-melanins. Tricyclazole, a specific inhibitor of reductase activity in the DHN-melanin pathway [20], provoked morphological and stress tolerance changes in the propagules differentiated by *M. robertsii* ARSEF 2575. Since the reductase enzymes involved in DHN-melanin synthesis are unique to this pathway, inhibiting these enzymes with tricyclazole leads to the accumulation of shunt products derived from the autoxidation of intermediate metabolites such as 1,3,6,8-THN that are converted into flaviolin [35], which serves as a viable method for inferring DHN-melanin formation [36]. This suggests that DHN-melanins might be present in the pigmented hyphal aggregates that the fungus differentiates in liquid culture, and that these dark pigments play an important role in MS formation and tolerance to both thermic and oxidative stress. Guaiacol, another inhibitor of DHN-melanin synthesis via a different mechanism to tricyclazole, possibly during melanin polymerization [26], also showed phenotypical and genic expression differences in the propagules obtained with its supplementation in the culture media. Thus, a hypothesis about the presence of DHN-melanin in this fungus is proposed.

Previous studies have established that *M. robertsii* is not capable of synthesizing DHN-melanins. Rangel et al. [37] reported that no scytalone dehydratase activity was found in cytoplasmic extracts of 4-day-old mycelium from *M. robertsii* ARSEF 2575. Moreover, Fang et al. [38] identified a laccase (MLAC1) in this fungus, which was postulated to be involved in conidial pigmentation, tolerance to abiotic stresses, and pathogenicity. However, conidial color formation has not been altered with different chemical agents that block specific metabolic pathways involved in melanization such as tricyclazole, kojic acid, and glufosinate ammonium. Finally, Chen et al. [29] characterized two polyketide synthase genes (designated as *Mrpks1* and *Mrpks2*) that showed homologies to those counterparts for the biosynthesis of heptaketide pigments and dihydroxynaphthalene (DHN)-melanins, respectively. In particular, in the *Mrpks1* cluster, a protein containing an EthD domain shares high similarity with the dehydratase AurZ, responsible for producing the red pigment aurofusarin in the plant pathogenic fungus *F. graminearum* [39]. Additionally, this cluster encompasses a gene encoding the laccase Mlac1 that shares similarity with the laccase Gip1 of *F. graminearum*, known for its involvement in pigment dimerization [40]. Conversely, the *Mrpks2* cluster is highly similar to the DHN-melanin biosynthesis gene clusters in both *Al. alternata* and *A. fumigatus* [29]. In this regard, Chen et al. [29] showed that culture pigmentation and the conidial wall structure were impaired by the disruption of *Mrpks1* but not *Mrpks2*, suggesting that non-melanin bis-naphthopyrone pigment/s are responsible for the spore color. In the present study, a downregulation of both *Mrthnr* and *Mrpks2* genes was observed in fungi harvested from the guaiacol-containing cultures, both of which might be implicated in the DHN pathway in *M. robertsii*. Notably, since *Mrpks2* is part of a cluster highly similar to the DHN-melanin biosynthesis gene clusters in both *Alternaria alternata* and *Aspergillus fumigatus* [29], therefore, while preliminary, this reduction in *Mrpks2* and the specific reductase of the DHN pathway gene expressions may complement the results on tricyclazole inhibition, reinforcing the hypothesis that DHN-melanins are present in the MS from *M. robertsii.* Furthermore, the upregulation of the *Mrpks1* gene suggest the activation of an alternative non-melanin pigmentation pathway in *M. robertsii* in compensation for the DHN pathway inhibition. In this regard, it would be interesting to study MS formation using melanin inhibition assays with the mutants *Mrpk1* and *Mrpks2* [29], as well as *Mrpks1*- and *Mrpks2*-overexpressed strains to fully understand the role of melanin and non-melanin pigments at different developmental stages. All this together leads us to establish that *M. robertsii* is capable of synthesizing different types of pigments according to different growth conditions and developmental stages, as has been described for other filamentous fungi [19]. In this sense, *Sporothrix schenckii* (Ophiostomataceae) is capable of producing DHN, DOPA, or pyomelanins according to different nutritional conditions and/or to the structures that the fungus differentiates [41,42,43].

A hyperoxidant state in a cell structure is considered to be a transient condition that needs to be battled to prevent harmful damage, which could lead to cell death [44]. Cell differentiation in eukaryotes, including fungi, has been proposed as a process triggered by hyperoxidant states, allowing cell protection from molecular oxygen. In some fungi, mycelial differentiation can induce sclerotial biogenesis [3]. In concordance with the results described in this work, tricyclazole at concentrations that inhibit melanin synthesis also inhibits the normal development of the sclerotia in *B. cinerea* [45] and *Sclerotinia sclerotiorum* [46]. Thus, it has been proposed that melanins may be key in contributing to structural rigidity in complex hyphal aggregates such as sclerotia [36]. Oxidative stress has been shown to be involved in scattered mycelium differentiation in the sclerotia of *Sclerotium rolfsii*, *S. minor*, *S. sclerotiorum*, and *Rhizoctonia solani* [3,47,48,49]. Oxidative stress’s role in MS differentiation has been predominantly examined in *M. rileyi*; initially, transcriptome analysis revealed an upregulation of antioxidant enzymes during MS formation [50]. Further investigations into culture optimization identified Fe^2+^ as a promoter of MS formation [13]. Notably, ferrous ions are recognized to elevate ROS levels via the Fenton reaction [3,50]. Additionally, the exogenous supplementation of menadione and hydrogen peroxide into the culture medium induced MS biogenesis, while the presence of ascorbic acid, acting as an ROS scavenger, inhibited MS formation [51]. It was found that *B. bassiana*’s MS had high peroxidase activity and peroxisomal proliferation during microsclerotial differentiation [16]. Moreover, several antioxidant response genes encoding for catalases and superoxide dismutases were induced in these propagules, as well as significantly higher levels of catalase and superoxide dismutase activities compared to those from scattered mycelia grown on glucose [16]. Melanin’s structure and its distinctive light absorption properties, particularly attributed to the presence of quinones and hydroquinones, serve to stabilize semiquinone free radicals within the polymer. Therefore, melanin stands out as one of the rare stable free radicals [36]. Consequently, it exhibits an absorptive capacity for other free radicals generated under environmental and/or internal stress conditions. Since lipid compounds are among one of the precursors of fungal melanins, the peroxidizable lipids available during the biogenesis of the MS in this fungus might be involved in the aggregation of the hyphae in a pseudoparenchymatous and melanized structure [5,52]. Ellil et al. [53,54] reported that lipid peroxidation and melanin formation are operating in parallel during the sclerotial biogenesis and maturation of the phytopathogen fungus *S. rolfsii.* In *B. bassiana*, pigment isolation and quantification from the microsclerotia and mycelia showed that the microsclerotial cells contained three times the amount of pigments of the control cells; thus, it was proposed that pigmentation occurs in immature MS to make the microsclerotia more insensitive to oxygen radicals, and thus, decreasing the ROS in the cell [16]. In the plant-pathogenic fungus *Verticillium dahliae*, melanized MS play a crucial role in its pathogenic cycle [55]. Additionally, the *Vayg1* gene from *V. dahliae*, a homolog for the *A. fumigatus* gene *Aayg1* involved at the first stages of DHN-melanin synthesis, was found to be upregulated during the development of melanized microsclerotia. This gene had been proposed to be key for DHN-melanin production and microsclerotial formation. Additionally, a *Vayg1* deletion mutant exhibited reduced virulence and oxidative stress resistance [56]. Also, the *VdMRTF1* gene (encoding a bZip transcription factor) was identified as a negative regulator of melanin production, microsclerotial development, the capability to eliminate endogenous ROS, and virulence in *V. dahliae* [57]. Concomitantly, microsclerotia from albino and dehydratase-deficient mutants of *V. dahliae* were unable to survive desiccation in soil. In contrast, wild-type microsclerotia and albino microsclerotia treated with scytalone to induce melanization entered a dormant state upon desiccation and resumed germination after 15 days of soil rehydration [58]. Our results indicate that propagules obtained in the presence of tricyclazole are less tolerant to desiccation and more susceptible to high temperatures and oxidative stress conditions. In this regard, Paixao et al. [30] reported that the pigmented MS of *M. robertsii* were more tolerant to UV and high temperature stress than non-pigmented pellets. Melanin production in fungi is closely tied to their ability to endure harsh environments [20]. These dark pigments shield cells from environmental stresses, and play a crucial role in thermoregulation. For instance, melanin provides protection to *C. neoformans* against temperature extremes, ensuring fungal resilience in both high and cold temperatures [59]. Similarly, melanin contributes to withstanding extreme temperature fluctuations in *Exophiala (Wangiella) dermatitidis* [60]. Additionally, the presence of melanin in the hyphae of *Phyllosticta capitalensis*, a widely distributed foliar endophyte, is attributed to the fungus’s survival in challenging conditions [61].

## 5. Conclusions

We report alterations in the differentiation from conidia to MS when *M. robertsii* is cultivated in the presence of DHN-type melanin inhibitors. Based on our findings, we infer that MS from *M. robertsii* ARSEF 2575 are capable of synthesizing DHN-type melanins. These dark pigments might play a crucial role in MS differentiation and their tolerance to both thermal and oxidative stresses. Additional experiments need to be conducted to fully understand the role of melanin and other non-melanin pigments at the different developmental stages of this fungus, in order to contribute to the development of new and optimized strategies to enhance fungal tolerance to abiotic stress conditions within the context of field biocontrol programs.

## Figures and Tables

**Figure 1 jof-09-01162-f001:**
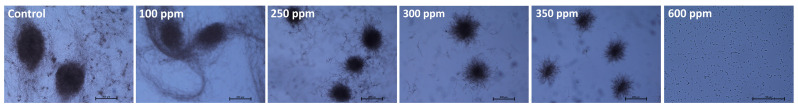
Hyphal aggregates of *Metarhizium robertsii* ARSEF 2575 produced in submerged liquid cultures unsupplemented and supplemented with tricyclazole in the concentration range between 100 and 600 ppm registered at 100× magnification using a light microscope.

**Figure 2 jof-09-01162-f002:**
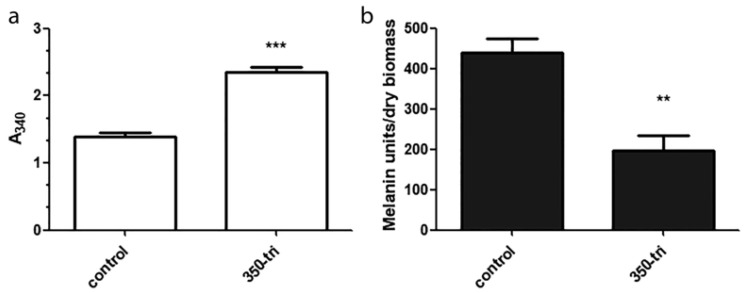
(**a**) Flaviolin accumulation measured by absorbance at 340 nm in media broth retrieved from 4-day-old cultures for MS production in *Metarhizium robertsii* (control) and cultures supplemented with 350 ppm tricyclazole (350-tri). (**b**) Melanin equivalents in *Metarhizium robertsii* MS (control) and 350-tri propagules. Values represent the mean of melanin equivalents/dry biomass ± SEM. Asterisks indicate significant differences (** *p* < 0.001, *** *p* < 0.0001).

**Figure 3 jof-09-01162-f003:**
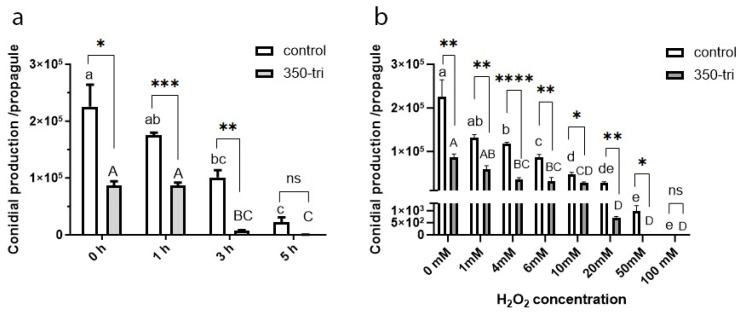
Reproductive (conidial-promoting) potential of *M. robertsii* differentiated propagules on a liquid culture unsupplemented (control) and supplemented with 350 ppm of tricyclazole (350-tri) in response to exposition to 45 °C (**a**) and hydrogen peroxide (**b**). Values indicate mean and SEM. At each exposure time, significant differences between propagules are shown with asterisks (* *p* < 0.05, ** *p* < 0.001, *** *p* < 0.0001, **** *p* < 0.00001), ns denotes no significant differences. For each condition (control: uppercase letters, 350-tri: lowercase letters), different letters indicate significant differences (*p* < 0.01).

**Figure 4 jof-09-01162-f004:**
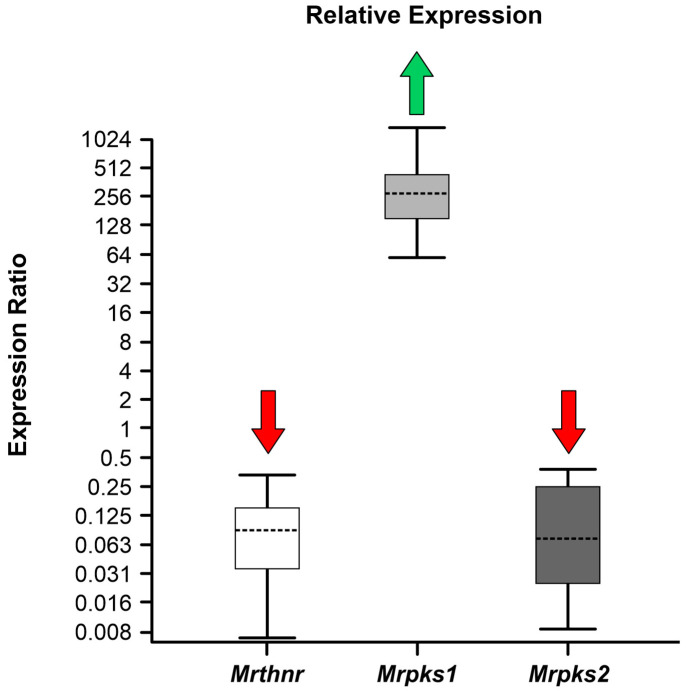
Boxplots showing gene expression analysis for *Mrthnr*, *Mrpks1*, and *Mrpks2* in *Metarhizium robertsii* ARSEF 2575 grown in media supplemented and unsupplemented with 75 ppm guaiacol. The box area encompasses 50% of all observations, the dotted line represents the sample median of three biological replicates, and the vertical bars represent the outer 50% of observations. Green arrows (up regulation) and red arrows (down regulation) indicate significant differences (*p* < 0.05).

**Table 1 jof-09-01162-t001:** Oligonucleotides used in this study.

Name	Forward Primer	Reverse Primer	Name/Function	Reference
*Mrtub*	TCGAGGGCTTCATGATGCTGC	CACGACCGAATCCGCATTCTG	Gamma-tubulin	This study
*Mrpks1*	CATTCCGCCTCTCTCATTGCC	TGTGCGGCGCATGATATGG	Polyketide synthase 1	Paixao et al. (2021) [30]
*Mrpks2*	CATCAGCGCCATCGGTTTAGAC	CGGGATAGGGATTGGTTTGTGG	Polyketide synthase 2	Paixao et al. (2021) [30]
*Mrthnr*	ATCAAGGCCGACATCACCAAGG	AATGTCCAGGTGGCCAAAGTGC	Putative 1,3,6,8-tetrahydroxynaphthalene reductase	This study

**Table 2 jof-09-01162-t002:** *Metarhizium robertsii* ARSEF 2575 liquid cultures supplemented with different melanin: pH of extracellular medium, dry biomass of the fungus, and production and diameters of its propagules. Asterisks indicate significant differences (** *p* < 0.001, *** *p* < 0.0001, **** *p* < 0.00001) between control and inhibitor-added media.

Inhibitor			Growth	pH	Dry Biomass (mg/mL)	Propagule Production (Units/mL) ×10^4^	Propagule Diameter (µm)
Inhibitor Name	Pathway Inhibited	Concentration (ppm)					
**Control**	**-**	0	+	5.1 ± 0.1	61 ± 4	1.6 ± 0.2	317 ± 14
**Bicyclopyrone**	**pyomelanin**	100	+	5.1 ± 0.1	55 ± 5	1.7 ± 0.3	339 ± 13
		350	+	5.0 ± 0.01	59 ± 7	1.8 ± 0.3	297 ± 9
		600	+	5.1 ± 0.1	59 ± 9	1.6 ± 0.3	308 ± 4
		1200	+	5.2 ± 0.2	57 ± 8	1.7 ± 0.2	337 ± 22
**Kojic Acid**	**DOPA**	100	+	5.4 ± 0.1	63 ± 7	1.6 ± 0.2	308 ± 7
		350	+	5.3 ± 0.1	66 ± 8	1.7 ± 0.3	306 ± 11
		600	+	5.5 ± 0.1	63 ± 6	1.5 ± 0.3	293 ± 13
		1200	+	5.5 ± 0.1	58 ± 7	2.0 ± 0.4	303 ± 8
**Tricyclazole**	**DHN**	100	+	3.4 ± 0.4 ***	66 ± 1	1.9 ± 0.5	256 ± 23
		350	+	3.8 ± 0.1 ***	60 ± 12	2.0 ± 0.2	100 ± 10 ***
		600	-	5.4 ± 0.2	7 ± 3 ***	-	-
**Guaiacol**	**DHN**	10	+	4.8 ± 0.1	63 ± 3	1.3 ± 0.2	291 ± 4
		35	+	4.9 ± 0.1	68 ± 2	4.9 ± 0.9 ****	203 ± 9 ***
		50	+	4.7 ± 0.1	78 ± 3	3.6 ± 0.3 ***	181 ± 8 ***
		75	+	4.9 ± 0.1	82 ± 5	3.4 ± 0.2 **	166 ± 9 ***
		100	-	5.2 ± 0.2	2 ± 1 ***	-	-

## Data Availability

Data are contained within the article.

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
