# Peer review of "Dark Pigments in Entomopathogenic Fungal Microsclerotia: Preliminary Evidence of a 1,8-Dihydroxynaphthalene-melanin-like Compound in Metarhizium robertsii"

_jof, 2023, doi:10.3390/jof9121162_

Round 1

Reviewer 1 Report

Comments and Suggestions for Authors

The manuscript submitted by Espín-Sánchez et al. reports a composition of dark pigments they identified in M. robertsii microsclerotia, which are promising propagules to be formulated as fungal insecticides. They characterized the fungal microsclerotia produced in the submerged cultures alone or supplemented with four types of melanin inhibitors at different concentrations. Their data showed that the inhibition of 1,8-dihydroxynaphthalene (DHN)-melanin biosynthetic pathway by tricyclazole and guaiacol resulted in the production of morphologically altered propagules, which were less tolerant to desiccation, heat, and oxidative stress than microsclerotia produced in the inhibitor-free culture. Therefore, they consider that the DHA melanin is a main type of melanin not only present in the fungal microsclerotia but also involved in the differentiation and stress tolerance of microsclerotia. Overall, the study was technically robust, and the data supports the conclusion. I recommend the manuscript to be considered for acceptance after a revision.

 Suggestions:

Inclusion of 75 ppm guaiacol in the submerged cultures resulted in transcriptional repression of two DHA biosynthesis-required genes but upregulation of the other DHA biosynthesis-required gene. Whether differential regulation of these genes in the chemical presence versus absence means the inhibition of the DHN biosynthetic pathway requires an expanded discussion by citing references on their functions. The current discussion is lengthy but focuses insufficiently on the mentioned issue.

The study tested four types of melanin inhibitors. Adding information on these inhibitors following the mentioned pathways in the Introduction section may help to understand the experimental design.

Comments on the Quality of English Language

The writing English is fine. 

Author Response

Thank you very much for taking the time to review this manuscript and for your valuable suggestions. In response to your first point, we have added a paragraph providing detailed information on the gene clusters and the functions of the protein-coding genes in other species with supporting references (lines 293-302). Regarding your second point, we have incorporated a paragraph into the Introduction section that provides additional information on the four types of melanin inhibitors, following the mentioned pathways (lines 57-66). This addition aims to improve the clarity and understanding of the experimental design. All incorporations are highlighted in red in the revised manuscript. 

Reviewer 2 Report

Comments and Suggestions for Authors

In this work, the authors studied the nature of microsclerotia pigmentation produced by Metarhizium robertsii after exposure to four melanin biosynthesis inhibitors. The work is interesting and presents information that can contribute to the greater development and use of entomopathogenic fungi in the management of agricultural pests.

The introduction is well written and presents relevant and up-to-date information on the topic. However, I would like the authors to consider the importance of including more information about the importance of their work associated with the end use of entomopathogenic fungi under field conditions for the management of agricultural pests.

The material and methods topic presents classic, robust, and widely reported methodologies in the literature. The results topic is stated clearly and coherently with the methodology described. The discussion addresses the main information contained in the results, using updated literature. However, despite the text being well written, I recommend that the authors revise it, as M. robertsii appears in many parts of the wrongly written text.

Author Response

Thank you for your time to review this manuscript, and for your positive feedback and constructive comments. As you suggested, we have included additional information in the Introduction section highlighting the importance of microsclerotia and its potential significance for the control of agricultural pests using entomopathogenic fungi under field conditions (lines 29-31; 38-41). All incorporations are highlighted in red in the revised manuscript. Also, the text has been thoroughly revised. We appreciate your keen observation, and the necessary corrections have been made to ensure clarity and accuracy.